# The IL-1 Family and Its Role in Atherosclerosis

**DOI:** 10.3390/ijms24010017

**Published:** 2022-12-20

**Authors:** Leticia González, Katherine Rivera, Marcelo E. Andia, Gonzalo Martínez Rodriguez

**Affiliations:** 1Centro de Imágenes Biomédicas—Departamento de Radiología, Escuela de Medicina, Pontificia Universidad Católica de Chile, Santiago 3580000, Chile; 2Instituto Milenio de Ingeniería e Inteligencia Artificial Para la Salud, iHEALTH, Pontificia Universidad Católica de Chile, Santiago 7820436, Chile; 3Programa de Doctorado en Ciencias Médicas, Facultad de Medicina, Pontificia Universidad Católica de Chile, Santiago 3580000, Chile; 4División de Enfermedades Cardiovasculares, Pontificia Universidad Católica de Chile, Santiago 3580000, Chile

**Keywords:** interleukin-1 family, inflammation, atherosclerosis, cardiovascular disease

## Abstract

The IL-1 superfamily of cytokines is a central regulator of immunity and inflammation. The family is composed of 11 cytokines (with agonist, antagonist, and anti-inflammatory properties) and 10 receptors, all tightly regulated through decoy receptor, receptor antagonists, and signaling inhibitors. Inflammation not only is an important physiological response against infection and injury but also plays a central role in atherosclerosis development. Several clinical association studies along with experimental studies have implicated the IL-1 superfamily of cytokines and its receptors in the pathogenesis of cardiovascular disease. Here, we summarize the key features of the IL-1 family, its role in immunity and disease, and how it helps shape the development of atherosclerosis.

## 1. Introduction

Though cardiovascular disease (CVD) remains one of the top causes of death worldwide, an epidemiological transition has been observed, where disease is no longer concentrated in industrialized countries, but increasingly affects individuals from less developed regions [1]. The classic assumption of the coronary patient—white middle-aged man with hypercholesterolemia and hypertension—is no longer the reality of the population currently affected. This stresses the importance of understanding the mechanisms governing the disease, to develop better-suited therapies and guidelines for the management of CVD burden.

The common underlying cause behind the manifestations of CVD disease is atherosclerosis. The chronic accumulation of lipids and immune cells in the vessel walls leads to the generation of the hallmark of the disease— the fibrolipidic plaque. The process starts with the accumulation of modified apolipoprotein B-containing lipoproteins in the intima layer of medium and large-sized arteries [2]. The triggered inflammatory process facilitates the expression of adhesion molecules on the endothelial cell lining of the vessels, leading to the recruitment of immune cells (primarily monocytes) from circulation. Upon infiltration, monocytes differentiate into macrophages, which express scavenger receptors and can engulf the modified lipoproteins, becoming foam cells [2]. The accumulation of intracellular cholesterol, due to constant influx, activates the stress-response machinery within foam cells, resulting in cell death [3]. Cell debris, foam cells, and extracellular lipids coalesce at the center of the plaque in the so-called necrotic core, which is kept contained by the formation of the fibrous cap—a structure composed of modified smooth muscle cells (SMC) and extracellular matrix proteins [4]. Neutrophils also significantly contribute to atherosclerosis progression, releasing inflammatory mediators and reactive oxygen species and contributing to plaque destabilization by releasing matrix proteinases [5].

The IL-1 family has proven to be one of the key drivers of the athero-inflammatory process, particularly through IL-1β and IL-18 activity. The aim of this review is to summarize the current understanding of this cytokine family and describe its role in atherosclerosis development, which might help explore new therapeutic avenues targeting these inflammatory mediators.

## 2. Systemic Inflammation and Cardiovascular Risk

The concept of traditional risk factors for atherosclerotic CVD has been established based on large-cohort studies, including the Framingham study [6]. However, recent research and changes in lifestyle have expanded the concept of what is currently considered a risk factor. The discovery of statins among other therapies has dramatically reduced the contribution of low-density lipoprotein (LDL) cholesterol to the burden of the disease [7]. Instead, triglyceride (TG) levels in circulation, often in the form of TG-rich lipoproteins, and low levels of high-density lipoprotein (HDL) are the main lipid abnormalities found in affected patients [8,9,10].

Despite the use of appropriate therapies to control risk factors, 42% of patients develop a recurrent ischemic event during the following 10 years after the first episode [11]. In the United States alone, 1 in 10 patients is hospitalized after an acute myocardial infarction (MI) within the following 30 days, with an estimated impact of USD 718 million per year [12]. This residual risk can be explained by multiple factors, including the inflammatory nature of atherosclerosis. Inflammatory biomarkers are currently used for risk stratification and therapy guidance in CVD patients [13]. High-sensitivity C-reactive protein (hsCRP) is an acute-phase reactant and a non-specific inflammatory marker that has been widely studied as a marker for systemic inflammation in risk stratification [13]. Several epidemiological studies have shown an association between hsCRP and future cardiovascular risk in “healthy” individuals [13,14,15]. Liuzzo et al. first reported the use of hsCRP, in combination with serum amyloid A, in predicting poor outcome in patients with severe unstable angina [16]. Later, Morrow et al. showed that elevated hsCRP in patients with unstable angina and non-ST elevation myocardial infarction (non-STEMI) correlated with increased 14-day mortality, alone and in combination with troponin T [17]. Furthermore, CRP was shown to be an independent predictor of cardiac risk and repeated coronary revascularization in a six-month follow-up study of unstable angina patients [18], and it was also associated with long-term risk of cardiac death, alone or in combination with troponin T [19]. Similar associations with long-term mortality were seen with CRP in non-STEMI patients undergoing aggressive revascularization strategies at early time points [20]. Finally, a significant association between hsCRP and future cardiovascular outcome has also been reported in patients with type 2 diabetes (T2D) and a recent acute coronary syndrome, despite treatment with statins and optimal glycemic control [21].

## 3. Athero-Inflammation, Cholesterol, and the NLRP3 Inflammasome

The important contribution of inflammation to atherosclerosis progression was already proposed by Rudolf Virchow in the 19th century; however, decades of research centered on the lipid hypothesis minimized this observation until recently [22]. Beyond the previously mentioned role of innate immunity (in the form of monocyte/macrophages and neutrophils), adaptive immunity has also been found to be involved in atherogenesis [23]. Indeed, the inflammatory process is involved in all stages of the disease: in the early phase through the infiltration of the vascular wall and the release of pro-inflammatory mediators that amplify the response, and at later stages, through the release of proteolytic enzymes by the immune cells, which contributes to plaque destabilization and the occurrence of acute coronary events [24]. The source of this low-grade vascular inflammation was early considered to be associated with infections, although no direct correlation between microbes and the development of atherosclerosis has been found [25]. Systemic inflammatory conditions—such as rheumatoid arthritis and psoriasis—are also associated with higher cardiovascular risk, indicating that systemic inflammation plays a crucial role in atherosclerosis development, independent of the source [26]. Wright et al. were the first to demonstrate that infectious agents are not required for the onset of atherosclerosis in germ-free apolipoprotein E mice, concluding that the high levels of cholesterol were sufficient to trigger the disease [27]. Instead, abundant research has shown that modified lipoproteins, as well as extracellular lipids in the form of cholesterol crystals (CCs), are known triggers of inflammation [28,29], establishing a connection between both the lipid and inflammatory hypothesis.

One of the ways by which CCs trigger inflammation is via activation of the nucleotide-binding oligomerization domain-like receptor, pyrin domain-containing 3 (NLRP3) inflammasome [30]. It has been suggested that, once phagocytosed, deficient clearance of CCs can induce lysosomal malfunction, resulting in the release of cathepsin B to the cytoplasm and the activation of the inflammasome [31,32]. Oxidized LDL has also been shown to activate the NLRP3 inflammasome through a similar mechanism, after interacting with the scavenger receptor CD36 [29].

Upon activation, a configurational change exposes the pyrin domain in NLRP3, allowing its interaction with the adaptor protein apoptosis-associated speck-like protein containing a caspase-recruitment domain (ASC) and the recruitment of procaspase-1, forming the inflammasome complex [33]. This interaction triggers the cleavage and activation of caspase-1 and then the processing of pro-IL-1β and pro-IL-18 into their active forms [34]. In the context of CVD, the activation of the NLRP3 inflammasome in macrophages, with the consequent release of IL-1β, has been shown to be implicated in the development of atherosclerosis and the clinical manifestations of the disease [30].

## 4. The IL-1 Family

### 4.1. History

The study of cytokines started in the 1940s with the active search for a fever mediator. Reports showed that supernatants of rabbit neutrophils were able to induce a rapid fever that was not due to contamination with endotoxin [35]. In 1974, Dinarello et al. described what they called “two distinct leukocytic pyrogens”, isolated from supernatants of neutrophil and monocyte cell cultures [36]. These two pyrogens differed in their molecular weight, isoelectric point, and capacity to induce sustained fever [36]. In 1977, the same group purified one of these pyrogens, showing that it had the ability to produce fever at a dose of 10 ng/kg [37]. The interleukin nomenclature was introduced in 1979, but the name IL-1β was not used until its cDNA was cloned in 1984 [38]. In the same year, a macrophage supernatant compound named “lymphocyte activating factor” was cloned in mice, revealing a second gene coding for IL-1 [39], and in the following year, human IL-1α was cloned [40].

In the early 1980s, there were reports of IL-1 inhibitory activity in the circulation of humans suffering from endotoxemia [41], which was then found in cultures of human monocytes [42] and the urine of patients with juvenile arthritis [43]. By the end of the decade, this inhibitory activity was attributed to a receptor antagonist of IL-1α and IL-1β [44]. In 1990, this component was identified as the IL-1 receptor antagonist (IL-1Ra) [45]. Several years later, the “IFNγ-inducing factor” was identified as IL-18 [46]. The remaining members of the IL-1 family were identified later, in the early 2000s [47].

### 4.2. Members of the IL-1 Family

#### 4.2.1. Cytokines

There are 11 members in the IL-1 family of cytokines (Table 1), which could play either an agonistic or antagonistic role in relation to their receptors, with one—IL-37—considered to be anti-inflammatory [35]. They are divided in subfamilies according to the length of the precursor and the propiece for each precursor (Figure 1). The IL-1 subfamily is formed by IL-1α, IL-1β, and IL-33 and has the longest propieces. The IL-18 subfamily is formed by IL-18 and IL-37, with a smaller propiece. The IL-36 subfamily is formed by IL-36α, IL-36β, IL-36γ, and IL-36 receptor antagonist (IL-36Ra), with the shortest propiece [35]. Evolution analysis showed that these cytokines coevolved and are present in all vertebrates, appearing about 420 million years ago, underpinning their key functional role [48]. With the exception of IL-1Ra, the members of the IL-1 family are mostly intracellular, since they lack a signal peptide and need to be cleaved to generate a mature, active cytokine [49]. They are found primarily in the cytoplasm as precursors containing the conserved AXD amino acid sequence: A, an aliphatic amino acid; X, any amino acid; D, for aspartic acid [49]. This conserved AXD motif is important for the proper folding of the IL-1 family of cytokines into their three-dimensional structure [50]. Nine amino acids from the AXD sequence, the N-terminus cleavage site, were found, which allows the proper folding of the cytokines [50]. This is also the site where caspase-1 cleaves IL-1β [51].

This ancient family of cytokines is known for playing a key role in the orchestration of both innate and adaptive immune responses (Figure 2). They have been reported to act on a wide variety of immune cells, triggering type 1, 2, and 3 responses [49,52,64]. IL-1α and β have been involved in emergency hematopoiesis, in situations where an increased number of hematopoietic cells are needed, such as aging, chemotherapy, and inflammation [52]. IL-18 is a key factor in the regulation of type 1 immunity, acting on NK cells, group 1 innate lymphoid cells (ILC1), and Th1 cells [52]. IL-33 is central in the modulation of type 2 immunity, adaptive immunity, and inflammation [49]. It also plays a key role in models of influenza virus and allergic lung inflammation, and single nucleotide polymorphisms in the genes coding for IL-33 are associated with asthma [65]. IL-1α and β have also been reported to regulate IL-17 production, which is involved in protection against bacterial and fungal infections, particularly in mucosal tissue [64,66]. For more in-depth information regarding the role of the IL-1 family of cytokines in immunity, please refer to the following reviews [49,51,52,56].

#### 4.2.2. Receptors

There are 10 members of the IL-1 family of receptors (Table 2) [67]. Like their ligand counterparts, they are found in all vertebrates and originate through gene duplication events, with some exceptions (e.g., IL-18R) [48]. They are characterized by an extracellular portion that presents Ig domains (usually three, with the exemption of IL-1R8 presenting only one) and the intracellular portion containing a Toll/IL-1 receptor (TIR) domain (except IL-1R2) [67]. The TIR domain allows for the interaction with specific TIR-containing adaptor proteins that are important for the activation of intracellular pathways, including: myeloid differentiation factor 88 (MyD88), TIR-domain-containing adapter-inducing interferon-β (TRIF), TRIF-related adaptor molecule (TRAM), sterile alpha- and armadillo-motif-containing protein (SARM), and TIR-domain-containing adapter protein (TIRAP/Mal) [68]. The active form of the receptors is a heterodimeric complex composed of a ligand-binding chain (IL-1R1, 2, 4, 5, and 6) and an accessory chain (IL1R 3, 3b, and 7) [67]. The proximity of the TIR domains to the two receptor chains initiates the intracellular signaling pathway. This signaling pathway most commonly involves the recruitment of MyD88 and IL-1R-associated kinase (IRAK), interaction with tumor necrosis factor (TNF)-α receptor-associated factor 6 (TRAF6), and finally, the activation of nuclear factor kappa B (NF-κB) [51].

Despite the shared use of MyD88 as an adaptor protein, the specificity of the IL-1 response is tightly controlled through the differential expression of both receptors and regulatory molecules. IL-1α and IL-1β are regulated by the expression of IL-1Ra and the decoy receptor IL-R2 [51]. A similar case is that of the IL-36 family, which is regulated by IL-36Ra [51]. IL-33 is regulated by cellular localization and by soluble receptors, while IL-18 activity is regulated by the IL-18 binding protein (IL-18BP) [67]. Another important regulatory molecule is IL-1R8 (also known as single Ig IL-1R-related molecule, SIGIRR) [70]. IL-1R8 is a decoy receptor mainly expressed by endothelial cells that can regulate the response to several IL-1 cytokines, including IL-1α, IL-1β, IL-33, and IL-18 [70]. IL-1R8 has also been reported to interfere with toll-like receptor signaling [70]. As key regulators of the host immunity, such a detailed system of regulation is required [72]. Failure to control processing and secretion of IL-1 cytokines results in a variety of syndromes characterized by fevers, rash, and other inflammatory manifestations, which will be discussed in the following sections.

### 4.3. Role in Disease

Autoimmune diseases are characterized by the dysregulation of the innate immune system, which becomes active in the absence of external threats. Many of these diseases are associated with episodes of systemic inflammation and fever, in which IL-1 cytokines play a role [53]. IL-1β has been implicated in the pathogenesis of rheumatoid arthritis (RA) [73], gout arthritis [74], ankylosing spondylitis (AS) [75], and adult-onset and systemic-onset juvenile idiopathic arthritis (Still’s disease) [47]. Treatment of these diseases with anakinra—an inhibitor of the IL-1R—or canakinumab—a monoclonal antibody against IL-1β—has been shown to effectively control inflammation, improving patients’ condition [53,76,77,78,79].

Excess IL-1 signaling is involved in some autoimmune hereditary syndromes such as Familial Mediterranean Fever (FMF), caused by a mutation in the MEFV gene, which codes for the pyrin protein [56]. Anakinra administration to FMF patients results in rapid symptom regression [56,80]. Colchicine, the standard treatment for FMF, is also a known blocker of NLRP3 inflammasome, which directly impacts IL-1β production [81]. Other syndromes where altered IL-1 signaling has been suggested to play a role, and in which anakinra treatment has resulted in positive results, are PAPA syndrome, hyperimmunoglobulinemia D periodic fever syndrome, TNF receptor-associated periodic syndrome, the non-hereditary Schnitzler syndrome [56], and cryopirinopathies [82]. A syndrome caused by a homozygous mutation in the gene coding for IL-1Ra has also been described, in which deficiency of IL-1Ra leads to severe inflammatory manifestations [83,84].

IL-1 cytokines are involved in maintaining the balance between tolerance to commensal microbiota and infectious agents [52]. High serum levels of IL-18 are associated with intestinal inflammation [52] and transgenic mice are more prone to colitis [85], while IL-18 inhibition is protective in models of inflammatory bowel disease (IBS) [86]. Inflammation and high levels of IL-18 have also been described in patients with Crohn’s disease [87,88]. IL-33 plays a role in IBS which varies depending on disease stage and inflammatory state [89]. In ulcerative colitis, blocking the IL-33 signaling pathway significantly reduced active disease [90]; however, IL-33-deficient mice are highly susceptible to colitis and colorectal cancer [91].

The IL-1 family of cytokines has been shown to be involved in neuroinflammation [53]. NLRP3 or caspase-1 deficient mice are protected from neuroinflammation and cognitive decline in models of Alzheimer’s disease [92]. In humans, *IL1A* allelic polymorphism leading to increased levels of IL-1α is associated with susceptibility to Alzheimer’s disease in some ethnic groups, due to promotion and secretion of the amyloid precursor protein [93]. Increased IL-1 also increases the risk of Parkinson’s, schizophrenia, some types of epilepsy, and febrile convulsions [94,95,96].

Finally, IL-1 cytokines have been implicated in the pathophysiology of CVD. As mentioned previously, inflammation is a key process in atherosclerosis development and is particularly critical in the transition to unstable plaques, which are associated with the clinical manifestations of the disease [2]. The role of the different subfamilies in atherosclerosis and CVD will be discussed in detail in the following sections.

## 5. IL-1 Subfamily

The IL-1 subfamily of cytokines is composed of IL-1α, IL-1β, IL-33, and IL-1Ra. IL-1α and IL-1β signal through IL-1R1, while IL-33 signals through IL-1R4 [47]. All three cytokines share IL-R3 as a co-receptor. The IL-1 subfamily is known to activate pro-inflammatory pathways, leading to the production of other cytokines and chemokines. IL-1Ra, on the contrary, specifically reduces the activity of both IL-1α and IL-1β [47]. IL-1R2 acts as a decoy receptor for both IL-1α and β, due to the similarities of the extracellular region of IL-1R1 [97]. The cytoplasmic region, however, is short and cannot signal. IL-1R2 can also interact with IL-1R3, allowing it to sequester the co-receptor as well [69]. A soluble, active form of IL-1R2 has also been described [98].

### 5.1. IL-1 Alpha

Evolutionary studies indicate that the IL-1α gene emerged from a duplication of the ancestral *IL-1β* gene [48]. The *IL-1α* gene consists of seven exons and six introns, with no TATA box in the promoter area, and generates a 31 kDa precursor protein [53]. The IL-1α precursor is constitutively expressed in epithelial cells, endothelial cells, and keratinocytes [53]. It can also be found in the surface of several cells, in particular monocytes and B cells, where it is called membrane IL-1α [99]. Membrane IL-1α can interact with and activate the IL-1R1 of adjacent cells, in a mechanism called juxtacrine [99].

IL-1α is considered a “dual-function” cytokine, since it can act both at the nucleus and cell membrane levels [51]. It contains a nuclear localization signal sequence in the precursor region, which allows it to induce the increased gene expression of other inflammatory agents, such as chemokine 8 [100]. The nuclear localization of IL-1α might also help control its pro-inflammatory properties. In the context of cell death, apoptotic signals trigger the localization of IL-1α to the nucleus, preventing inflammation [101]. In contrast, during necrosis, IL-1α moves from nucleus to cytosol, contributing to the highly inflammatory milieu of necrotic cell lysates [101]. Due to this property, IL-1α is considered a DAMP, activating several inflammatory reactions when interacting with its receptor [102]. Interestingly, the precursor of IL-1α is active, being able to engage its target receptor when released in the context of necrotic cell death, suggesting a key role in sterile inflammation [103]. This role of the IL-1α precursor has earned it the role of an “alarmin” [103].

### 5.2. IL-1 Beta

IL-1β is one of the most studied members of this family. The *IL-1β* gene has a structure similar to the one coding IL-1α, except for the presence of a TATA box near the promoter [53]. Contrary to IL-1α, IL-1β is not expressed in a constitutive manner in healthy individuals and it is only produced by a limited number of cell types including monocytes/macrophages and dendritic cells [53]. IL-1β is expressed as a 31 KDa inactive precursor, often in response to an insult (microbial products, other cytokines, and even IL-1α and IL-1β), in a process called priming [104]. In a second step, processing of the IL-1β precursor is triggered, rendering a fully matured 17 KDa protein able to engage the IL-1R [104]. Caspase-1 is required for this second step, which in turn is activated by the inflammasome [105]. The following release of mature IL-1β seems to involve externalization through vesicles [106], but secretion mediated by pyroptosis, a type of cell death associated with inflammation, has been described in monocytes/macrophages [107].

Production of mature IL-1β has also been described independent of caspase-1. Mature IL-1β has been detected in turpentine-induced necrosis in caspase-1 KO mice [108]. Neutrophil proteinase 3 (PR3) has been shown to process pro-IL-1β in a model of acute neutrophil-dominated arthritis lacking caspase-1 [109]. PR3 can also process extracellular IL-1β precursors, and similar activity has been described for granzyme, matrix metalloprotease 9, and neutrophil elastase [72]. Upon interacting with the IL-R1, IL-1β results in the activation of NK-κB, through the IRAK pathway, resulting in the production of downstream inflammatory mediators [105]. As such, IL-1β plays a central role in the local inflammatory response, mediating the production of IL-6, which can in turn act systemically, activating, among others, acute phase response proteins such as CRP [105].

### 5.3. IL-33

IL-33 was identified as a member of the IL-1 family in 2005 by Schmitz et al. [110]. Like IL-1α, it is considered a dual-function protein: it works in the nucleus regulating transcription and as an extracellular cytokine, upon release due to stress [52]. The gene contains seven exons, generating a 31 KDa protein with a nuclear localization sequence in the N-terminal domain and a receptor-binding sequence in the C-terminal [54]. IL-33 has been reported to be expressed in a constitutive manner in the nucleus of epithelial cells and the vascular endothelium, where it forms a complex with chromatin [55,111]. In conditions of necrotic cell death, the chromatin-IL-33 complex is disrupted, allowing for the release of IL-33 to the extracellular space as an alarmin signal [112]. Given this property, IL-33 has been shown to be involved in priming for allergic responses in basophils, mast cells, and eosinophils, among others [113]. It has also been shown to be prominent in the pathogenesis of several allergic conditions including asthma and atopic dermatitis [114,115].

IL-33 signals through the complex between IL-1R3 and its unique receptor suppressor of tumorigenesis 2 (ST2), also known as IL-1R4 [110,116]. ST2 is expressed in a transmembrane and a soluble form [110]. Both forms of this receptor are significantly expressed in aortic and coronary endothelial cells as well as immune cells [117,118]. ST2 activation by IL-33 involves the recruitment of MyD88, IRAK-1 and 4, and TRAF6, with the resulting activation of NF-κB and mitogen-activated protein kinases (MAPK) [119]. Several mechanisms have been described that regulate IL-33 bioavailability and activity. The interaction of IL-33 with soluble ST2 and the nuclear localization have both been described as involved in the neutralization of this cytokine [119]. Phosphorylation of transmembrane ST2 at S442, which results in the internalization and further degradation of the receptor by the proteasome, and inhibition of IL-33 signaling by the interaction of TIR domain of IL-1R8 with the IL-33 receptor complex have been reported as well [119,120].

### 5.4. IL-1 Receptor Antagonist

IL-1Ra was first described as a molecule with IL-1 inhibitory activity in the supernatant of human monocyte cultures in 1985 [56]. The *IL-1RN* gene can be expressed as four different protein forms, including IL-1Ra, due to alternative splicing [53]. IL-1Ra is a natural antagonist for IL-1α and IL-1β, since it competes with the IL-1 molecules for IL-1R1 [35]. Two forms of IL-1Ra have been described: secretory and intracellular [53]. The secretory form interacts with IL-1R1, limiting IL-1α and IL-1β activities. Even though IL-Ra has a similar affinity for IL-1R1 when compared with IL-1α and IL-1β, large molar quantities are required to block IL-1 activity, which might be explained by the high amount of IL-1R1 expressed in most cells [121]. The mechanism of action of the other three intracellular forms is less clear but it might involve intranuclear competition with IL-1α and a non-IL-1R1-dependent inhibition of intracellular signaling [72]. Dying cells have been shown to release intracellular IL-Ra, which can then bind IL-1R1 [56].

The important role of IL-1Ra has been demonstrated in KO animal models. IL-1Ra KO mice display increased inflammatory responses and develop joint and skin inflammation as well as vasculitis, according to their genetic background [56]. Severe clinical manifestations of inflammation were also reported in children with mutations in the IL-1RN gene, displaying pustular rash, joint swelling, and elevated CRP, among other manifestations [83,84].

### 5.5. Role of the IL-1 Family in Atherosclerosis

IL-1 cytokines are involved in the whole process of atherosclerosis development (Figure 3). Both IL-1α and IL-1β have been reported to be expressed in atherosclerotic plaque [122]. The principal source of these cytokines in the plaque are macrophages [122]. The local production of both cytokines can impair the normal function of endothelial cells, inducing oxidative stress, the production of procoagulant factors and impairing vasodilation, all of which can accelerate atherothrombosis [123]. Locally, several mediators of atherosclerosis pathogenesis have been shown to induce IL-1 cytokine expression and release. Oxidized LDL can interact with toll-like receptors and CD36, leading to the expression of IL-1α and IL-1β [29,124]. Necrosis and apoptosis, which occur at the height of the transition towards mature plaques, have also been reported to lead to IL-1 cytokine release [125].

The role of IL-1α and IL-1β has also been explored in animal models of atherogenesis. ApoE KO mice lacking IL-1β develop significantly smaller lesions compared to control mice [126]. Similar results were also seen when a neutralizing antibody against IL-1β was used in the same animal model [127]. Smaller lesions were also reported in apoE KO mice receiving either IL-1α or IL-1β-deficient bone marrow (BM) compared to controls, highlighting the importance of hematopoietic-derived IL-1 [128]. In a similar study, Kamari et al. showed that IL-1α deficiency seems to confer a higher degree of protection against atherosclerosis development [129]. Protection against fatty streak formation, the earliest step in lesion development, was also reported in C57BL6 mice lacking IL-1α [129]. Interestingly, combined deficiency of IL-1α and IL-1β does not result in additive protection [128]. Reduced expression of IL-1Ra in apoE KO mice resulted in larger lesions at early stages with significantly more macrophages, pointing to a role in modulating plaque composition [130]. Administration of recombinant IL-Ra [131] and overexpression of IL-Ra in LDL receptor (LDLR) KO mice resulted in reduced atheroma formation as well [132]. In the same line, IL-1R1 deficiency reduces atherosclerotic burden in the aortic root of apoE KO mice, reducing outward remodeling and increasing plaque stability [133]. Interestingly, atheroprotective effects of IL-1β have been described in later stages of disease development. Treatment of apoE KO mice with advanced lesions (long-term western-diet feeding) with an IL-1β antibody resulted in a significant reduction in SMC and collagen content, but increased the number of macrophages in the fibrous cap [134]. These changes did not affect lesion size but inhibited outward remodeling, which has been described as beneficial [134]. It is likely that the interplay between monocytes and SMC is a dynamic process, with different responses according to the stage of plaque development. Overall, it seems that NLRP3 inflammasome activation promotes a phenotypic change in SMC, fostering vulnerability and resulting in rupture-prone plaques [135].

CCs are another local mediator that can lead to IL-1 cytokine production in atherosclerotic plaque. CCs can activate the NLRP3 inflammasome by inducing the release of the lysosomal protease cathepsin to the cytoplasm, which in turn activates the NLRP3 inflammasome. LDLR KO mice transplanted with NLRP3, ASC, or IL-1β-deficient bone marrow developed smaller plaques when challenged with an atherogenic diet [28]. Similar results were observed with systemic or bone-marrow deficiency of caspase-1/11 in both apoE [136,137] and LDLR KO mice [138]. However, apoE KO mice lacking NLRP3, ASC, or caspase-1 did not show effects on atherosclerotic plaque size, plaque stability, or macrophage infiltration, stressing the importance of biological and experimental context [139]. Conversely, administration of a NLRP3 inflammasome inhibitor for 4 weeks resulted in a reduction in plaque burden in apoE KO mice [140]. Expression of the NLRP3 inflammasome has been reported in human plaques correlating with the severity of coronary artery disease (CAD) [141,142], and also in circulating monocytes of acute coronary syndrome (ACS) patients, compared to controls [143].

Clonal hematopoiesis, a key feature of aging, also involves IL-1 cytokines [52]. The expansion of the ten-eleven-translocation-2 (TET2) mutant hematopoietic clones is suggested to be an important driver of atherosclerosis [144]. Bone marrow transplantation of atherosclerosis-prone mice with TET2-deficient bone marrow resulted in an increase in plaque development, which was associated with increases in NLRP3-mediated IL-1β production in TET2-deficient macrophages [145]. In humans, carriers of clonal hematopoiesis of indeterminate potential (CHIP) experience a 2-fold increase in the risk of coronary artery disease and a 4-fold increase in early-onset myocardial infarction, compared to non-carriers [145].

In patients, genetic differences in the IL-1 gene cluster, which are associated with higher levels of pro-inflammatory cytokines, are associated with the presence of CAD, identified through invasive angiography [146]. It also increased the effect of oxidized phospholipids and lipoprotein(a) on cardiovascular events [146]. Canakinumab treatment of patients with stable CAD, with a history of previous MI and increased hsCRP, resulted in a reduction in cardiovascular events, highlighting the contribution of the IL-1 pathway to cardiovascular risk [147]. IL-1β concentrations have also been reported to be associated with an increased risk of death in patients with heart failure [148]. Similarly, in patients with acute MI, IL-1β levels at admission were independently associated with all-cause mortality, and high levels of IL-1β were associated with cardiovascular death and major adverse cardiovascular events (MACE) [149]. A meta-analysis carried out by Herder et al. showed that circulating levels of IL-1Ra were positively associated with CVD risk, based on the data of six population-based cohort studies [150]. This effect of IL-1Ra levels on CVD risk did not impact, however, overall risk prediction based on classical risk factors [150]. Treatment of STEMI patients with anakinra, on the other hand, did not have the effect of preventing recurrent ischemic events, but it might prevent new-onset heart failure [151]. Finally, a randomized, phase II trial looking at the effect of Xilonix—an anti-IL-1α antibody—on neointimal hyperplasia and post-procedural inflammation in the superficial femoral artery after angioplasty, showed a trend towards a reduction in restenosis [152].

## 6. IL-18 Subfamily

The IL-18 subfamily comprises IL-18 and IL-37, which have opposite effects (Figure 3). On one side, IL-18 is a potent pro-inflammatory cytokine originally described as an IFN-γ inducing factor by Th1 cells [46,153]. In contrast, IL-37 has been described as an anti-inflammatory cytokine due to its functions as a non-specific inhibitor of inflammation [154]. It was initially described as a cytokine that prevented IL-18-mediated inflammation [59]. Consequently, there is active research looking at the potential role of IL-37 in fighting several types of inflammatory diseases, including atherosclerosis [154,155].

### 6.1. IL-18

IL-18 is a pleiotropic pro-inflammatory cytokine generated because of proteolytic cleavage of pro-IL-18 by intracellular caspase-1 in the NLRP3 inflammasome [57,156]. The active form is secreted extracellularly, where it can exert several biological functions by binding to IL-18R, a heterodimeric complex consisting of the ligand-binding chain termed IL-1R5, and the co-receptor chain or signal-transducing chain termed IL-1R7 [58]. IL-18R is highly expressed in NK cells, driving its differentiation and activation. Nonetheless, several observations have also documented its expression in Th1 cells, B cells, CD4  +  NKT cells, mast cells, and basophils [157].

IL-18 activates intracellular signaling pathways by first recruiting MyD88 and activating NF-κB and MAPK pathways mainly to induce IFN-γ production for host defense against intracellular pathogen infection, which is augmented by IL-12 [46,158]. In fact, differentiation of naive T cells to the Th1 phenotype is synergistically induced by IL-12 and IL-18 [159]. In addition, there is a well-described role for IL-18 driving the Th2 response in the absence of IL-12 but in the presence of IL-2, involving the stimulation of NK cells, NKT cells, and even established Th1 cells to produce IL-3, IL-9, and IL-13 [160,161]. Thus, IL-18 stimulation can determine different responses that initiate and promote host defense and inflammation [57].

IL-18 is biologically and structurally related to IL-1β and is constitutively expressed in most cell types including human peripheral blood mononuclear cells (PBMCs) [162], macrophages [163], DCs [164], and vascular endothelial cells [165]. To prevent uncontrolled immune responses, IL-18 is regulated by the soluble IL-18BP, which can neutralize IL-18 activity due to a higher affinity for IL-18 compared to the IL-1R5, thus suppressing IFN-γ production and limiting cell responses [166,167,168]. Interestingly, recent evidence suggests IL-37 acts as a negative regulator or antagonist member of IL-18 by binding to IL-1R5 and blocking downstream signaling [169].

### 6.2. IL-37

IL-37 is a unique member of the IL-1 family that suppresses both innate and acquired immunity to maintain an anti-inflammatory state and can act as an anti-inflammatory alarmin [59,60,170]. The tertiary structure of the IL-37 precursor is closely related to the IL-18 precursor and the intron–exon borders of the IL-18 and IL-37 genes suggest a close association [171]. In fact, IL-37 is produced as a precursor, which is also cleaved by caspase-1 to generate its mature form [60]. Thus, IL-37 is mainly induced in an inflammatory context and IL-1β, IL-18, TNF-α, IFN-γ, and TGF-β can increase its synthesis [51], resulting in a compensatory inhibition of the production of pro-inflammatory cytokines [154]. There are five isoforms of IL-37, named from a to e—the IL-37b isoform is mainly detected in immune cells from peripheral blood [172]. The mechanism of action of IL-37 is based on the formation of a complex with IL-1R5 and IL-1R8 on the cell surface, which suppresses the phosphorylation of several inflammatory kinases such as mTOR [154,173]. In fact, IL-37 suppresses the production of IL-1β mediated by the NLRP3 and AIM2 inflammasomes, and the production of IL-18 by the NLRP3 inflammasome [174,175]. Moreover, IL-37 can also translocate to the nucleus and block Smad3 activation, resulting in anti-inflammatory effect due to the suppression of NF-κB and MAPK [51,174]. Interestingly, the anti-inflammatory properties of IL-18BP described above for IL-18 are lost when IL-37 levels increase, probably because IL-18BP also binds to IL-37 and consequently inhibits the anti-inflammatory effects of IL-37 itself, suggesting that the net IL-37 anti-inflammatory effect depends on both IL-18R and IL-18BP [154,169].

In humans, IL-37 is constitutively expressed in several different tissues and cells, such as blood monocytes, tissue macrophages, plasma cells, and T cells, which may help in the maintenance of the immune homeostasis [176]. While there is no known mouse homolog of human IL-37, transgenic mice expressing human IL-37b are protected from acute inflammation [177,178]. Importantly, altered IL-37 expression in the serum has been found in patients with different inflammatory diseases, suggesting that this cytokine is a potential tool for treating inflammation-driven diseases [154].

### 6.3. Role of the IL-18 Family in Atherosclerosis

The administration of exogenous IL-18 to athero-prone mice resulted in larger atherosclerotic lesions [179]. IFN-γ plays a key role in atherosclerotic immune cell recruitment, foam cell formation, and plaque progression and stability [180,181,182]. Accordingly, it is believed that the IL-18 pro-atherosclerotic effect is mediated by IFN-γ, which acts on lesion cells including macrophages, endothelial cells, and vascular SMC (VSMC) to accelerate plaque formation [165,179]. As expected, these effects are abolished in IFN-γ-deficient mice [179]. Interestingly, the proatherogenic effect of IL-18 can also occur in the absence of T cells—IFN-γ secreted by macrophages, NK cells, and VSMC has been shown to be sufficient for disease progression [183]. On the other hand, a recent study showed that induction of cell adhesion molecules by recombinant IL-18 via activation of NF-κB might be a crucial event in initiating atherosclerotic lesion development [184]. In fact, exogenous recombinant IL-18 administration significantly upregulates CD36 and NF-κB, while a NF-κB inhibitor blocked IL-18 signaling, restoring plaque stability [185,186]. This suggests a CD36 and NF-κB crosstalk involved in the progression and destabilization of atherosclerotic plaques. Paradoxically, IL-18-deficient mice have shown lipid abnormalities and increased lipid deposition in the arterial wall, as well as enhanced macrophage and Th17 infiltrate, IL-23-producing VSMC and macrophages, and a morphology resembling unstable plaques [187,188]. In this model, Th17 rather than Th1 produced IFN-γ, and IL-18 might not be needed for IFN-γ production in these lesions [188]. Interestingly, total cholesterol levels were positively regulated by IL-18, while HDL cholesterol may reduce IL-18BP levels [159]. This is relevant since it has been demonstrated that the inhibition of IL-18 activity by IL-18BP prevented fatty streak development and slowed down the progression of advanced atherosclerotic plaques in mice [189]. In addition, it was recently shown that both IL-18 and Na-Cl co-transporter (NCC) are necessary to coordinate the IL-18R cell signaling in atherosclerosis, rising a novel pathway for signaling in atherogenesis and plaque vulnerability involving the IL-18/IL-18R dyad [190].

Increased levels of the novel anti-inflammatory IL-37 have been reported in several human disease conditions [155]. Specifically, a previous study showed IL-37 expression in the foam-like cells of human atherosclerotic coronary and carotid artery plaques [191]. The imbalance of M1 and M2 macrophage polarization is directly associated with the progression of atherosclerotic disease, and a recent study showed a higher amount of M1 macrophages and less IL-37 expression in calcific aortic valves, compared to normal valves [192]. In addition, macrophage-specific IL-37 expression led to reduced plaque development and decreased systemic inflammation in atherosclerosis-prone mice [193]. These observations suggest that IL-37 could shift macrophage polarization from the pro-inflammatory M1 phenotype to the anti-inflammatory M2 phenotype, which is particularly abundant in stable zones of plaque and asymptomatic lesions [194]. Interestingly, IL-37 has also been implicated in the regulation of cholesterol homeostasis and prevention of the initiation of atherosclerotic plaque development through decreased NF-κB signaling and reduced MCP-1 production in IL-37-Tg mice, suggesting a role in avoiding mononuclear cell infiltration [195,196].

A recent study showed that mice overexpressing IL-37 exhibited significant improvements in their atherosclerotic burden and plaque stability through differentiation of the T helper cell anti-inflammatory phenotype, decreasing matrix metalloproteinase (MMP)-2/13-mediated degradation of collagen and inhibition of VSMCs apoptosis [197]. Furthermore, it was recently shown that exogenous IL-37 resulted in a reduction in atherosclerotic plaque size through a decrease in Th1 and Th17 cells and an increase in Treg cells in mice [198]. In addition, it was shown that IL-37 may also attenuate atherosclerosis and atherosclerosis-related diseases through the inhibition of DC activation [199,200]. Moreover, recombinant IL-37 enhances vascular endothelial function by increasing the bioavailability of nitric oxide, and improves systemic insulin sensitivity and glucose tolerance, effects that can potentially prevent atherosclerosis development [201].

Blankenberg et al. were the first to show that serum levels of IL-18, in patients with documented CAD (stable and unstable angina), significantly predicted death by cardiovascular causes during follow-up [202]. Levels of IL-18 are also significantly elevated in patients with previous MI compared to controls, and they are significantly associated with coronary plaque area [203]. IL-18 levels have been shown to be prospectively predictive of cardiovascular events in healthy individuals, in nested case-control studies [204,205,206]. These results were further confirmed by a large prospective study in healthy men carried out by Jefferis et al., which also included a systematic review and meta-analysis of all available prospective data [204].

Taken together, these data strongly suggest that the IL-18 subfamily is a key mediator in the atherosclerotic pathogenesis as well as a potential biomarker.

## 7. IL-36 Subfamily

The IL-36 subfamily includes three proteins with agonistic activity—IL-36α, IL-36β, and IL-36γ-, one with antagonistic activity—IL-36Ra, and one with potential receptor antagonist activity—IL-38 (Figure 3) [51]. IL-36R is expressed in monocytes, myeloid dendritic cells (mDCs), and monocyte-derived dendritic cells (MDDCs) from normal mice to respond to IL-36, which is primarily secreted in the skin by keratinocytes, but also has been shown to be secreted in T cells, lung, and gut cells [207]. The IL-36 subfamily can be regulated by different cells and inflammatory components, such as IFN-γ, IL-1α, IL-22, IL-17A, and TNF-α [207]. In recent years, the IL-36 subfamily has generated great interest because of its dysregulation in inflammatory diseases [208].

### 7.1. IL-36 α, β, γ

IL-36α, IL-36β, and IL-36γ exert strong pro-inflammatory effects and share 21–37% sequence homology with IL-1 and IL-1Ra [61]. IL-36 cytokines are unlikely to be secreted by a caspase-1-dependent mechanism (as is the case for IL-1β and IL-18) because they lack a caspase cleavage site, suggesting that IL-36 cytokines adopt an alternative secretion mechanism, putatively, extracellularly [62,209]. Specifically, IL-36α needs both elastase and Cathepsin G for Lys3 and Ala4 cleavage and resulting activation [210]. In contrast, Cathepsin G cleaves IL-36β at residue Arg5, while Cathepsin S secreted by endothelial cells can cleave IL-36γ between Glut17 and Ser18 residues [210,211].

The active forms are obtained after removal of the nine amino acids preceding the AXD site, which allows them to bind IL-1R6, which dimerizes with IL-1R3 to activate NF-κB and MAPK transcription factors in a MyD88-dependent manner [50,51,62]. IL-36 cytokine signaling pathways appear to be tissue, cell, and context dependent [212]. For example, while primary human keratinocytes stimulated with IL-36α, IL-36β, or IL-36γ induced the expression of inflammatory cytokines [210], IL-36γ-treated endothelial cells result in IL-8, CCL2, CCL20, vascular cell adhesion molecule 1 (VCAM-1), and intercellular adhesion molecule (ICAM-1) upregulation [213]. Moreover, monocytes and DCs cultured with IL-36 resulted in an increased production of IL-6 and IL-12, suggesting that IL-36 plays an important role in the innate immune response [207,214].

### 7.2. IL-36Ra

IL-36Ra is an anti-inflammatory cytokine that shares 54% homology with IL-1Ra [61]. IL-36Ra is also synthesized as an inactive progenitor and its antagonistic function requires its cleavage at a methionine residue situated at the N-terminus by neutrophil-derived elastase to assume its active form [215]. IL-36Ra binds to IL-1R6 with a higher affinity than the agonists but does not lead to IL-1R3 recruitment and the consequent signaling, thereby competing and inhibiting the activity of IL-36α, IL-36β, and IL-36γ in a dose-dependent manner [50,216]. Specifically, it has been shown that IL-36Ra antagonizes IL-36α in mouse models that overexpress this agonist [217]. IL-36Ra is highly expressed in both immune and non-immune cells, and mice with IL-36Ra deficiency showed reduced weight gain and metabolic dysfunction [51,217].

### 7.3. IL-38

IL-38 is an IL-36 antagonist sharing similar characteristics with IL-36Ra and IL-1Ra [63,216]. Like IL-36 cytokines, IL-38 is secreted as a precursor and does not contain caspase-1 cleavage sites, yet its activation process is not completely understood [218]. The mechanism by which IL-38 exerts its anti-inflammatory actions also remains to be clarified; however, it was shown that the cleaved form of IL-38 can downregulate IL-6 expression by binding IL-1R9 (IL-1 receptor accessory protein l, IL-1RAPL1) and, subsequently, inhibiting the Jun N-terminal kinase (JNK) pathway [71]. It has also been proposed that IL-38 could instead bind IL-1R6 and IL-1R1, resulting in a reduction in IL-8 production [216]. Despite these data, further studies are needed to delineate the specific molecular mechanism of IL-38.

### 7.4. Role of the IL-36 Subfamily in Atherosclerosis

Experimental models have demonstrated that IL-36γ can exert atherosclerosis-promoting effects by augmenting macrophage foam cell formation and uptake of oxidized low-density lipoproteins through increased expression of the scavenger receptor CD36 in mice [219]. On the other hand, IL-36Ra inhibited the development of atherosclerosis by acting on the NLRP3 inflammasome [220]. From a clinical standpoint, it was recently shown that serum levels of IL-36 were significantly higher in patients with CAD compared to controls, and these levels correlated with the serum concentrations of TNF-α, IL-6, IL-32, and coronary artery stenosis [221]. Although these novel findings suggest a possible role for the IL-36 family in CAD, more data are eagerly awaited.

## 8. Conclusions

For several decades, the pathophysiology of atherosclerosis was thought to be mediated and driven exclusively by alterations in the metabolism and accumulation of lipids. However, current knowledge points to a strong influence of inflammation on atherosclerosis development, with both local and systemic effects. Moreover, there is a clinical need to address the residual inflammatory risk currently left untreated by available therapies. Among the different inflammatory mediators involved in atherosclerosis onset and progression, the IL-1 superfamily of cytokines plays a key role. Several research articles and clinical studies have provided insight into the contribution of the most well-known members of this superfamily to atherosclerosis development and its clinical manifestations (Table 3). Recent research has primed the potential benefit of non-specific (i.e., Colchicine [222]) and specific (i.e., Canakinumab [147]) IL-1 inhibitors to reduce cardiovascular events. However, the IL-1 superfamily also plays a critical role in host immunity, which might directly hamper the adoption of IL-1 blockers as potential treatments, as recent trials have raised concerns about the effects on non-cardiovascular events (such as infections or cancers) secondary to these drugs. Therefore, the future challenge is to optimize the net benefit between reducing athero-inflammation without affecting immune defenses. Targeting specific inflammatory pathways that are overrepresented in atherosclerosis or potentiating immunomodulatory cytokines—such as IL-37—might help accomplish this long-awaited goal.

## Figures and Tables

**Figure 1 ijms-24-00017-f001:**
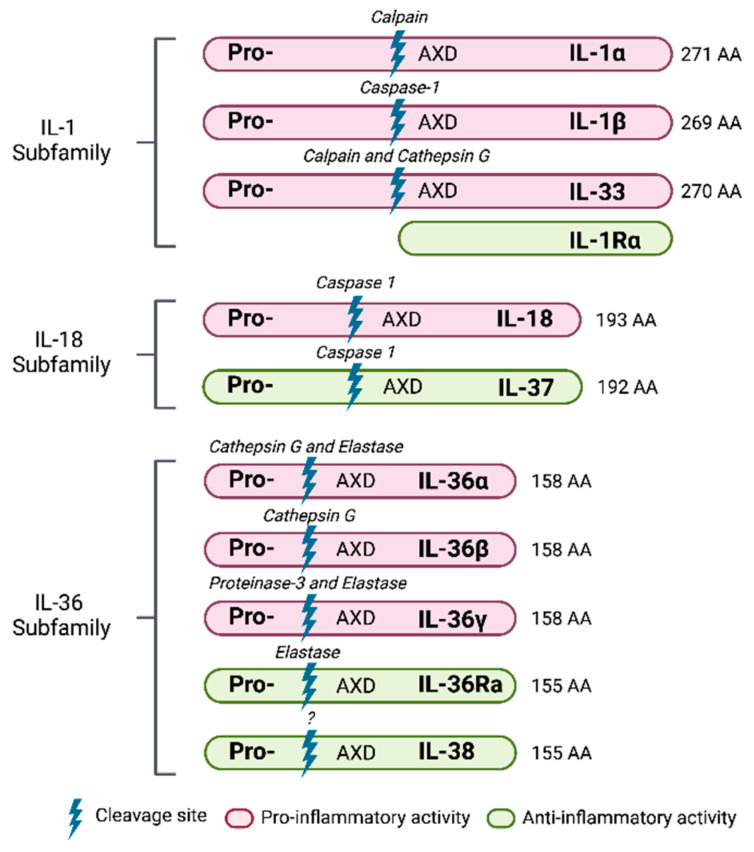
IL-1 superfamily of cytokines. Schematic representation of the three subfamilies, processing enzymes, and main role.

**Figure 2 ijms-24-00017-f002:**
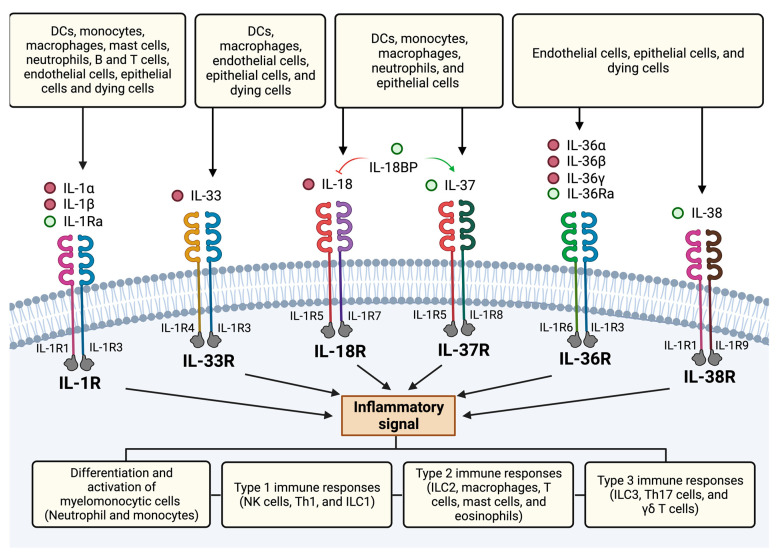
IL-1 superfamily in innate and adaptive immunity. IL-1 cytokines exert their effects on several cells of both the innate and adaptive immune system, triggering type 1, 2, and 3 immune responses. DCs = dendritic cells, NK = natural killer cells, Th1 = Th helper 1 cells, ILC = innate lymphoid cells, Th17 = Th helper 17 cells.

**Figure 3 ijms-24-00017-f003:**
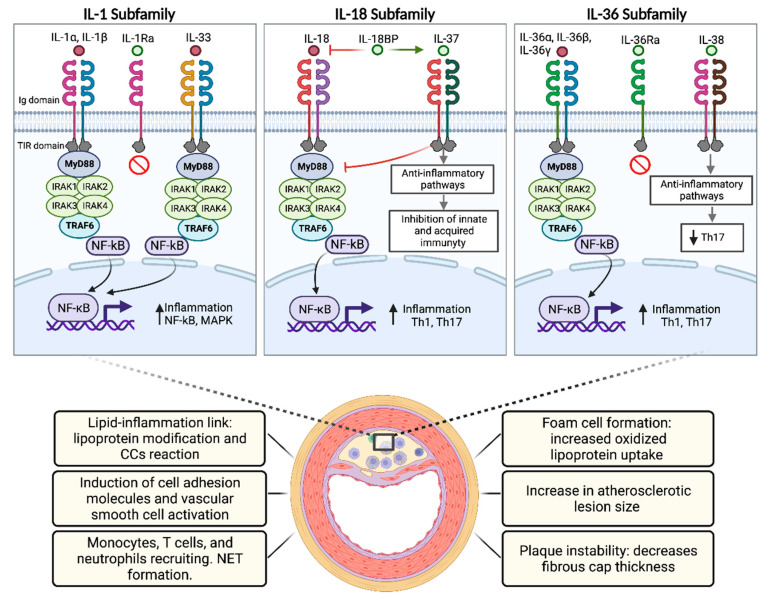
Role of the IL-1 family in atherosclerosis. Common signaling pathway for IL-1 family cytokines, which bind to IL-1R family members, recruiting MyD88 and IRAK and resulting in the activation of NF-κB and MAPK and then promoting the transcription of several atherosclerotic pro-inflammatory genes. CCs: Cholesterol Crystals; NET: Neutrophil Extracellular Traps.

**Table 1 ijms-24-00017-t001:** IL-1 family of cytokines.

Cytokine Name	Alternative Name	Subfamily	Main Role	References
IL-1α	IL-1F1	IL-1	Pro-inflammatory	[52,53]
IL-1β	IL-1F2	IL-1	Pro-inflammatory	[52,53]
IL-33	IL-1F11	IL-1	Pro-inflammatory, type 2 immunity	[52,54,55]
IL-Ra	IL-1F3	IL-1	Antagonist of IL-1α and IL-1β	[53,56]
IL-18	IL-1F4	IL-18	Pro-inflammatory, Type 1 immunity	[57,58]
IL-37	IL-1F7	IL-18	Anti-inflammatory	[59,60]
IL-36Rα	IL-1F5	IL-36	Antagonist of IL-36α, β, γ	[61,62]
IL-36 α, β, γ	IL-1F6, 8, 9	IL-36	Pro-inflammatory	[61,62]
IL-38	IL-F10	IL-36	Anti-inflammatory	[63]

**Table 2 ijms-24-00017-t002:** IL-1 family of receptors.

Receptor	Alternative Name	Role	Ligands	References
IL-1R1	IL-1RI	IL-1 binding receptor	IL-1 α, IL-1β, and IL-1Ra	[49,67,69]
IL-1R2	IL-1RII	IL-1 decoy receptor	IL-1β	[49,67,69]
IL-1R3	IL-1RAcP	IL-1 co-receptor	IL-1 α, IL-1β, IL-33, IL-36	[49,67,69]
IL-1R4	ST2/T1	IL-33 binding receptor	IL-33	[49,67,69]
IL-1R5	IL-18Rα/IL-1Rrp1	IL-18 binding receptor	IL-18	[49,67,69]
IL-1R6	IL-36R/IL-1Rrp2	IL-36 binding receptor	IL-36	[49,67,69]
IL-1R7	IL-18Rβ	IL-18 co-receptor	IL-18	[49,67,69]
IL-1R8	TIR8/SIGIRR	IL-37 co-receptor/decoy receptor	IL-37	[69,70]
IL-1R9	APL/TIGIRR-2/IL-1RAPL	IL-38 co-receptor	IL-38	[69,71]
IL-1R10	TIGIRR, TIGIRR-1/IL-1RAPL2	Orphan receptor	-	[69]

**Table 3 ijms-24-00017-t003:** IL-1 family of cytokines and its role in atherosclerosis.

Cytokine	Animal Studies	Human Studies	References
IL-1α	(i) Deficiency in mice confers protection against disease, even at early stages. (ii) Bone marrow deficiency in mice results in small lesions and reduction in inflammation.	(i) Treatment with Xilonix results in a reduction in restenosis in the femoral artery after angioplasty.	[128,129,152]
IL-1β	(i) Deficiency or neutralization in mice results in smaller lesions. (ii) Bone marrow deficiency in mice results in smaller lesions. (iii) In advanced lesions, it affects VSMC phenotype. (iv) Inhibition of outward remodeling and promotion of plaque instability. (v) Increased levels are produced by macrophages in a clonal hematopoiesis-influenced atherosclerotic model.	(i) IL-1β concentration is associated with increased risk of death in HF patients. (ii) IL-1β levels at admission are independently associated with all-cause mortality in patients with acute MI. (iii) High levels are associated with cardiovascular death and MACE. (iv) Canakinumab administration to stable CAD patients results in reduction in cardiovascular events.	[126,127,128,134,135,145,147,148,149]
IL-1Ra	(i) Reduced expression in mice results in larger lesions and increased macrophage content at early stages. (ii) Recombinant IL-1Ra administration or overexpression affects lipoprotein metabolism and foam cell formation.	(i) Circulating levels are positively associated with CVD risk, without affecting overall risk prediction. (ii) Treatment of STEMI patients with anakinra might prevent new-onset HF.	[130,131,132,150,151]
IL-18	(i) Exogenous administration to mice results in larger plaques. (ii) Recombinant IL-18 administration upregulates CD36 and modulates NF-κB, inducing atherosclerosis. (iii) Pro-atherogenic effect is mediated by IFN-γ expression. (iv) Deficiency results in lipid abnormalities, lipid deposition and changes in morphology associated with unstable plaques. (v) Inhibition by IL-18BP prevents fatty streak development and slows down disease progression.	(i) Serum levels in CAD patients significantly predict CV death. (ii) Circulating levels are increased in patients with previous MI and are associated with coronary plaque area. (iii) IL-18 levels are prospectively predictive of CV events in healthy individuals.	[179,183,184,185,189,202,203,204,206]
IL-37	(i) Macrophage expression leads to reduced plaque development and decreased systemic inflammation. (ii) Regulation of cholesterol homeostasis and prevention of atherosclerosis (IL-37 transgenic mice model). (iii) Overexpression reduces atherosclerosis burden and improves stability through decreased collagen degradation and VSMC apoptosis. (iv) Exogenous administration increases Treg cells and decreases Th1 and Th17 cells, reducing plaque size. (v) Inhibition of DC activation could also mediate anti-atherogenic effect.	(i) Expression has been reported in human atherosclerotic plaques.	[191,193,195,196,197,198,199,200]
IL-36 subfamily	(i) IL-36γ promotes foam cell formation by increasing CD36-mediated oxidized lipoprotein uptake in mice. (ii) IL-36Ra attenuates atherosclerosis development through inhibition of the NLRP3 inflammasome in mice.	(i) IL-36 serum levels are increased in CAD patients, and they correlate with inflammatory cytokines and coronary artery stenosis.	[219,220,221]

CAD: Coronary Artery Disease; CV: Cardiovascular; CVD: Cardiovascular Disease; DC: Dendritic Cell; HF: Heart Failure; MACE: Major Adverse Cardiovascular Events; MI: Myocardial Infarction; NLRP3: Nucleotide-binding Oligomerization Domain-like Receptor, Pyrin Domain-containing 3; STEMI: ST elevation myocardial infarction; VSMC: Vascular Smooth Muscle Cells.

## Data Availability

Not applicable.

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
