# Peer review of "The IL-1 Family and Its Role in Atherosclerosis"

_ijms, 2022, doi:10.3390/ijms24010017_

Round 1

Reviewer 1 Report

This review manuscript by Gonzalez at al. summarizes the current knowledge about the role of IL1 family members in inflammatory diseases and atherosclerosis.  The authors deeply researched this  them; overall the manuscript is well written and contains large amount of information. However, several major and minor issues should be addressed before to publication.

Major

1.     While the title of the review article is “ The IL1 family and its role in atherosclerosis” rather a modest description about the role of IL1 family members in this disease is provided. As currently presented the review appears to describe the role of IL1 family cytokines in inflammatory diseases. It would be helpful for the reader if the information describing the role of IL1 family members in other immune and inflammatory diseases will be better integrated to what is known in atherosclerosis, including discussion of similarities/differences of IL1 family-regulated inflammatory mechanisms and extrapolation of described in other diseases mechanisms to atherosclerosis.

2.     Some recent publications by Garry Owens group suggested stage dependent and, perhaps, cell type dependent role of IL1 signaling in atherosclerosis. It would be important to describe and discuss it in the review

3.     It would be helpful to highlight in conclusion what could be next important research directions/unanswered questions in the field of IL1 family cytokines in atherosclerosis.

Minor

1.     Table 1. It would be helpful to add extra column with reference numbers

2.     Adaptive immunity instead of adaptative immunity should be used

3.     Fig 2. Effector cell types could be listed instead or in addition to “type 1, type 2 immune response”.

4.     Figure 3 and Figure 4 (which is labeled as Figure 1 as well ) have identical panels, simply Fig4 has an additional image of aorta.

Author Response

Reviewer 1

This review manuscript by Gonzalez at al. summarizes the current knowledge about the role of IL1 family members in inflammatory diseases and atherosclerosis.  The authors deeply researched this  them; overall the manuscript is well written and contains large amount of information. However, several major and minor issues should be addressed before to publication.

Major

  1. While the title of the review article is “ The IL1 family and its role in atherosclerosis” rather a modest description about the role of IL1 family members in this disease is provided. As currently presented the review appears to describe the role of IL1 family cytokines in inflammatory diseases. It would be helpful for the reader if the information describing the role of IL1 family members in other immune and inflammatory diseases will be better integrated to what is known in atherosclerosis, including discussion of similarities/differences of IL1 family-regulated inflammatory mechanisms and extrapolation of described in other diseases mechanisms to atherosclerosis.

A: We thank the reviewer for this comment. We have reordered the manuscript to highlight better the role of IL1 family in atherosclerosis. A relevant new table, which summarizes findings relating each cytokine with atherosclerosis has been added, in order to better address this topic. This table was also suggested by the other reviewer.

  1. Some recent publications by Garry Owens group suggested stage dependent and, perhaps, cell type dependent role of IL1 signaling in atherosclerosis. It would be important to describe and discuss it in the review.

A: We appreciate this comment. Recent research by this group has been added to the manuscript and discuss accordingly, in conjunction with new findings on the role of cytokines on vascular smooth cell regulation. Page 16, para 1.

  1. It would be helpful to highlight in conclusion what could be next important research directions/unanswered questions in the field of IL1 family cytokines in atherosclerosis.

 A: To the authors, the main challenge is balancing the inherent risk of targeting key inflammatory mediators in terms of host defenses with the potential benefit on atherosclerosis. This reflection has been more clearly added to the final conclusion. Although many other unanswered question remain on this topic, we believe that a more general statement better concludes the manuscript.

Minor

  1. Table 1. It would be helpful to add extra column with reference numbers
  2. Adaptive immunity instead of adaptative immunity should be used
  3. Fig 2. Effector cell types could be listed instead or in addition to “type 1, type 2 immune response”.
  4. Figure 3 and Figure 4 (which is labeled as Figure 1 as well ) have identical panels, simply Fig4 has an additional image of aorta.

A: All these have been corrected.

Reviewer 2 Report

The review is very comprehensive. It is supported by recent, extensive and relevant bibliography.

The paper gives an extensive overview of IL1 superfamily role in the regulation of inflammation and immunity. Other important aspect of this review is the demonstration of current body of evidence that associates mechanisms of action and disease outcomes. From my point of view this is a very interesting paper that will be very useful for physicians and researchers.   

However, the text is quite massive and does not facilitate reading comprehension. It even blurs the main objective and purpose of the review well stated in title.

IL1 relationship with CDV diseases is the main focus of the review.

Therefore, a Table/block diagram summarizing the relationship of IL1 subfamily relevant cytokines with outcomes as reported on investigations and whether these studies were conducted in animal models or are clinical studies, would be an added value. This will improve the reader’s awareness of how superfamily of IL1 cytokines are important to understand the wide range of CVD manifestations and severity.

Check mistyping

Fig 2 – top blocks. Substitute dyring by dying

Ln 702 – Figure 1 should be Fig. 4

Author Response

Reviewer 2

  1. The review is very comprehensive. It is supported by recent, extensive and relevant bibliography.

The paper gives an extensive overview of IL1 superfamily role in the regulation of inflammation and immunity. Other important aspect of this review is the demonstration of current body of evidence that associates mechanisms of action and disease outcomes. From my point of view this is a very interesting paper that will be very useful for physicians and researchers.   

However, the text is quite massive and does not facilitate reading comprehension. It even blurs the main objective and purpose of the review well stated in title.

IL1 relationship with CDV diseases is the main focus of the review.

Therefore, a Table/block diagram summarizing the relationship of IL1 subfamily relevant cytokines with outcomes as reported on investigations and whether these studies were conducted in animal models or are clinical studies, would be an added value. This will improve the reader’s awareness of how superfamily of IL1 cytokines are important to understand the wide range of CVD manifestations and severity.

A: We appreciate this comment from the reviewer. Indeed, this is a very extensive topic. We have reordered the manuscript to better highlight the role of the IL1 family in CVD disease. The section on the role of IL1 family in innate and adaptive immunity has been removed and readers are referred to relevant articles.

The table summarizing cytokines and outcomes has been added to the manuscript.

  1. Check mistyping

Fig 2 – top blocks. Substitute dyring by dying

Ln 702 – Figure 1 should be Fig. 4

A: All these have been corrected.
